# Development of an easy-to-use questionnaire assessing critical care nursing competence in Japan: A cross-sectional study

**Masatoshi Okumura**[1]*, **Tomonori Ishigaki**[2], **Kazunao Mori**[3], **Yoshihiro Fujiwara**[1]

**1** Department of Anesthesiology, Aichi Medical University, Nagakute, Aichi, Japan, **2** Department of Business Administration, Nanzan University, Nagoya, Aichi, Japan, **3** Division of Nursing, Aichi Medical University Hospital, Nagakute, Aichi, Japan

\* okuchin0518er@gmail.com

## Abstract

### Background

Critical care nurses need a high level of medical competence, especially with regard to patient safety. There are several tools to measure general and critical care nursing competence, but the usability of these tools is inadequate because they include large numbers of questions. To maintain quality and safety in intensive care units (ICUs), it is necessary to be able to easily measure and evaluate critical care nursing competence. The purpose of this study was to develop an easy-to-use questionnaire assessing critical care nursing competence related to patient safety.

### Methods

A cross-sectional, descriptive, explorative study was designed to collect data from nurses working in six ICUs in tertiary hospitals in Japan. Data were collected from August 2017 to December 2018. The Critical Care Nursing Competence Questionnaire for Patient Safety (C3Q-safety) is a 22-item instrument designed to assess nursing competence related to patient safety in ICUs. Items were developed based on previous work related to critical care nursing competence and were adjusted based on a pilot study.

### Results

A total of 211 nurses working in ICUs participated in this study. Through descriptive statistics and factor analysis, the number of questions was reduced from 24 to 22. The C3Q-safety had four factors: decision making, collaboration, nursing intervention, and principles of nursing care. Cronbach's alpha ranged from 0.73 to 0.83. The four factors showed positive correlations with each other (0.47 to 0.72). Nurses licensed as certified nurses in intensive care and those with longer ICU work experience showed significantly higher scores on all four factors.

**Data Availability Statement:** All relevant data are within the paper and its Supporting Information files.

**Funding:** The authors received no specific funding for this work.

**Competing interests:** The authors have declared that no competing interests exist.

## Conclusions

We developed an easy-to-use questionnaire to assess critical care nursing competence related to patient safety. The C3Q-safety was able to detect four areas of competence. The C3Q-safety will make it possible to easily measure critical care nursing competence and can be utilized for efficient education.

## Introduction

Critically ill patients need to receive not only expert care supported by in-depth knowledge and a high level skill, but also care that ensures patient safety. Therefore, critical care nursing requires a high level of competence, which is an ongoing challenge.

The term "competence" is a widely used concept for all types of professionals. Competence is also an essential concept in nursing, but the definition of this concept is vague [1, 2]. Critical care nursing competence comprises both clinical and professional competence [3, 4]. Clinical competence can be divided into three domains: principles of nursing care, clinical guidelines, and nursing interventions. Professional competence can be divided into four domains: ethical activity, decision making, development work, and collaboration. From a different viewpoint, critical care nursing competence consists of four domains: knowledge base, skill base, attitude and value base, and experience base [3]. DeGrande et al.'s (2018) integrative review of the literature noted that clinical and professional competence may overlap and that managing situations, decision making, and teamwork are important in developing professional competence.

In terms of the medical environment in Japan, as of 2016, there were 1033 certified nurses (CNs) in intensive care and 177 certified nurse specialists (CNSs) in critical care nursing among 805,708 full-time registered nurses (RNs) in the country [5, 6]. There are several ways to become an RN in Japan, but most RNs receive nursing education at a four-year college or a three-year training school and then pass the national nursing examination [6]. RNs are licensed nationally, whereas certified nursing licenses are issued by the Japanese Nursing Association (JNA). To become a CN in intensive care, RNs with more than five years of practical experience must complete a 6-month educational program at an educational institution and then pass an examination administered by the JNA. To become a CNS in critical care nursing, an RNs with more than three years of experience in a specialized field must complete a master's course at a graduate school and then pass an examination administered by the JNA. There are 653 intensive care units (ICUs) in Japan, with a median of seven beds per ICU [7]. The Japanese Ministry of Health, Labour and Welfare requires ICUs to have at least one nurse per two patients, but specific requirements such as level of qualification and ability are not defined for nurses working in ICUs by the Ministry, the JNA, or individual hospitals. Therefore, nurses can work in ICUs even as first-year RNs who have not participated in standardized training courses on topics such as advanced cardiovascular life support or fundamental critical care support. In other words, nurses in Japan are compelled to work in ICUs, where advanced skills are required, regardless of whether they have the appropriate skills or not.

To date, three measurement scales for critical care nursing competence have been published: the Critical Care Competency Assessment instrument, the Self-Assessment Competence Tool, and the Intensive and Critical Care Nursing Competence Scale (ICCN-CS-1) [4, 8, 9]. Because these instruments include a large number of items, these three scales cannot be administered frequently or easily. The ideal questionnaire should be usable by more people in usual clinical settings. Additionally, if the questionnaire can be included in the Plan-Do-

Study-Act cycle, the quality of health care can be improved. Therefore, the following four points must be considered for a questionnaire seeking to measure critical care nursing competence: What does the questionnaire measure? Can nurses complete the questionnaire without difficulty? Can the questionnaire be evaluated easily? From the evaluation, what kinds of training can we provide for quality improvement?

In summary, although ICU nurses are expected to provide particularly high-quality and safe nursing care, not all of them receive appropriate training on care. Furthermore, because there are no questionnaires measuring critical care nursing competence that are easy to use in clinical settings, education based on evaluation is currently difficult to accomplish. The main aim of the present study was to develop an easy-to-use self-assessment questionnaire that can be used to measure critical care nursing competence, focusing especially on patient safety.

## Materials and methods

### Design

A cross-sectional survey was carried out to validate and assess an original self-report questionnaire, the Critical Care Nursing Competence Questionnaire for Patient Safety (C3Q-safety). The study was conducted in three phases: instrument preparation, pilot study, and field study.

### Instrument

The C3Q-safety is a scale designed to detect nursing competence related to patient safety in ICUs. The initial 24 questions were derived from the nurse competencies of the American Association of Colleges of Nursing, the clinical ladder of the JNA, and face-to-face and written interviews with ICU physicians, ICU nurse practitioners, and CNs in intensive care. Each question was scored on a five-point Likert scale, with higher scores corresponding to better achievement. A pilot study was conducted with a sample of 49 ICU nurses. These results were discussed among the authors and considered in the development of the final version of the C3Q-safety.

### Sample and setting

Data were collected using self-report questionnaires administered to nurses working in the ICUs of four tertiary hospitals and two university hospitals in Japan from August 2017 to December 2018. The participants received an explanation about this study, and those who agreed to participate completed the questionnaire.

### Procedure

We used self-assessment questionnaires in paper form, including items on critical care nursing competence and demographic characteristics. Demographic data included sex, age, number of years working as a nurse, number of years working in a critical care department, possession of a CN in intensive care or CNS in critical care nursing license from the JNA, and level on the nursing clinical ladder in Japan.

### Analysis

SPSS, Version 17.0 was used to analyze the data. The demographic data and the questions on nursing competence were evaluated using descriptive statistics. The underlying structure of the C3Q-safety was explored using exploratory factor analysis. The principal factor method was used to extract the factors, followed by oblique rotation of factors using promax rotation. The number of factors to be retained was guided by Kaiser's criterion (eigenvalues $\geq 1$),

inspection of the screeplot, the cumulative contribution rate, and interpretation of the content of each factor. We used *t*-tests and one-way analysis of variance to examine the relationships between demographic characteristics and nursing competence. The α level was set at 0.05 for statistical significance. Bonferroni's correction was performed to maintain an overall type 1 error rate of 0.05 in multiple comparisons.

### Ethical considerations

This study was approved by the Ethical Committee of Aichi Medical University, Japan (ethical code: 2017-M036, 2018-M008). All of the participants were given written and verbal descriptions of the study, and their anonymity and confidentiality throughout the study were guaranteed. Participation in the study was voluntary, and returning the questionnaires was considered informed consent.

## Results

### Participants' demographic characteristics

Of 293 questionnaires distributed to ICU nurses, 211 were returned (response rate = 72.0%). Table 1 shows the participants' demographic characteristics. The study participants were primarily women (n = 164, 77.7%). The most common age category was 20–29 years (n = 125, 59.2%). Years working as an RN ranged from 1 to 26 (mean = 7.6, standard deviation [SD] = 6.3). Years working as an ICU nurse ranged from 1 to 17 (mean = 4.0, SD = 3.1). Level on the JNA's clinical ladder ranged from 0 to 4 (median = 2, interquartile range: 1–3), and level 2 was the most common level among study participants (n = 56, 27.7%). A total of 14 participants were licensed as CNs in intensive care or CNSs in critical care nursing by the JNA (6.6%).

### Item analysis of the C3Q-safety

Table 2 shows the results of an item analysis of the 24 items initially included in the scale. The mean values on the items ranged from 3.52 to 4.52, with SDs ranging from 0.59 to 0.83. Items 2 and 3 showed a ceiling effect, but we did not exclude these items because their scores showed a normal distribution. None of the items showed a floor effect.

### Factor analysis of the C3Q-safety

Exploratory factor analysis was conducted on the 24 items to examine the factor structure. Principal factor analysis was used. Five factors were extracted, explaining a total of 56.8% of the variance. Item 12 was excluded because of low communality, bringing the number of items to 23. Assuming a four- or five-factor structure, we conducted factor analysis with promax rotation on the remaining 23 items to examine construct validity. Item 24 showed low communality here and was excluded, resulting in a total of 22 items. The 22-item questionnaire had a four-factor structure. The variance contribution ratios of the four factors were 36.78%, 7.07%, 6.34%, and 4.92%. The cumulative contribution ratio was 55.1%. Table 3 shows the factor structure of the 22-item questionnaire. Based on Aari et al.'s classification of critical nursing competence areas and the content of the questionnaire items, the factors were named as follows: decision making (Factor 1, seven items), collaboration (Factor 2, five items), nursing intervention (Factor 3, five items), and principles of nursing care (Factor 4, five items). Decision making and collaboration are elements of professional competence, whereas nursing intervention and principles of nursing care are elements of clinical competence. The Cronbach's alpha coefficients, related to internal reliability, for the items on decision making, collaboration, nursing intervention, and principles of nursing care were 0.83, 0.80, 0.73, and 0.79,

**Table 1. Demographic characteristics of ICU nurses (n = 211).**

| Variable | | n | % |
|---|---|---|---|
| Sex | | | |
| | Male | 47 | 22.3 |
| | Female | 164 | 77.7 |
| Age (years) | | | |
| | 20–29 | 125 | 59.2 |
| | 30–39 | 64 | 30.3 |
| | ≥ 40 | 22 | 10.4 |
| Total nursing experience (years) | | | |
| | < 1 | 19 | 9.0 |
| | ≥ 1, < 5 | 90 | 42.7 |
| | ≥ 5, < 10 | 45 | 21.3 |
| | ≥ 10 | 57 | 27.0 |
| | Mean = 7.6 (SD = 6.3), Median = 5 (IQR: 3 to 11) | | |
| ICU nursing experience (years) | | | |
| | < 1 | 38 | 18 |
| | ≥ 1, < 5 | 132 | 62.6 |
| | ≥ 5, < 10 | 30 | 14.2 |
| | ≥ 10 | 11 | 5.2 |
| | Mean = 4.0 (SD = 3.1), Median = 3 (IQR: 2 to 5) | | |
| Clinical ladder level | | | |
| | 0 | 48 | 23.8 |
| | 1 | 44 | 21.9 |
| | 2 | 56 | 27.7 |
| | 3 | 34 | 16.8 |
| | 4 | 20 | 9.9 |
| | Median = 2 (IQR: 1 to 3) | | |
| CN in intensive care/CNS in critical care nursing | | | |
| | Yes | 14 | 6.6 |
| | No | 197 | 93.4 |

ICU: intensive care unit, SD: standard deviation, IQR: interquartile range, CN: certified nurse, CNS: certified nurse specialist

respectively (Table 3). The four factors showed positive correlations with each other (Table 4). Decision making had an especially strong positive correlation with all of the other factors.

## Multiple comparisons between demographic characteristics and nursing competence

Table 5 shows the results of the multiple comparison analysis between demographic characteristics and nursing competence areas. The total score on the four nursing competence areas (range: 4–20 points) differed significantly by whether the participant was licensed as a CN in intensive care/CNS in critical care nursing, level on the clinical ladder, length of nursing experience, and length of ICU nursing experience. Those licensed as CNs in intensive care/CNSs in critical care nursing scored significantly higher than did non-certified nurses on all four competences. The higher a nurse's level on the clinical ladder, the higher the nurse's score on decision making (Factor 1), collaboration (Factor 2), and nursing intervention (Factor 3), and these differences were significant (Factor 1: $p < 0.001$; Factor 2: $p < 0.05$; Factor 3: $p < 0.001$).

**Table 2. ICU nurse competence score on the C3Q-safety (n = 211).**

| | Questionnaire item | Mean (SD) |
|---|---|---|
| 1 | Before a patient enters the ICU, you have prepared the items needed for the patient's care. | 4.30 (0.59) |
| 2 | You begin patient monitoring within 3 minutes of the patient arriving in the ICU. | 4.52 (0.66) |
| 3 | You begin physical examinations within 5 minutes of the patient arriving in the ICU. | 4.26 (0.78) |
| 4 | You execute the doctor's orders on time (within 5 minutes). | 3.52 (0.81) |
| 5 | You conduct physical assessments using the ABC (airway, breathing, circulation) approach. | 3.75 (0.82) |
| 6 | When you notice abnormal vital signs or specific physical findings, you tell a doctor or other nurses. | 4.18 (0.59) |
| 7 | You prioritize monitoring and execution of the doctor's orders according to the patient's condition. | 4.08 (0.70) |
| 8 | You frequently communicate with ICU nurse co-workers. | 4.27 (0.65) |
| 9 | You frequently communicate with other medical staff (e.g., pharmacists, physical therapists, clinical engineering technicians). | 3.79 (0.83) |
| 10 | You share your anxiety with others. | 3.79 (0.78) |
| 11 | You repeat oral instructions aloud. | 4.22 (0.74) |
| 12 | You do not say anything irrelevant when examining a patient. | 3.90 (0.80) |
| 13 | You can recognize your role and act accordingly. | 4.02 (0.63) |
| 14 | You can fulfill your role. | 3.60 (0.74) |
| 15 | You clean up empty medicines and organize complicated infusion lines. | 4.18 (0.63) |
| 16 | You respond flexibly to changed orders. | 3.91 (0.71) |
| 17 | After finishing a task, you independently find a new task. | 3.85 (0.67) |
| 18 | You solve problems in clinical practice. | 3.84 (0.70) |
| 19 | You take thoughtful care of patients (e.g., smiling, touching). | 4.24 (0.68) |
| 20 | You value what the patient says. | 4.08 (0.67) |
| 21 | You actively provide patients with nursing that leads to their recovery. | 3.83 (0.64) |
| 22 | You listen attentively to the patient's family's anxiety. | 4.01 (0.64) |
| 23 | You provide nursing equally to all patients. | 3.78 (0.74) |
| 24 | You take standard precautions when you touch a patient. | 4.02 (0.66) |

ICU: intensive care unit, C3Q-safety: Critical Care Nursing Competence Questionnaire for Patient Safety, SD: standard deviation

Longer nursing experience was linked to higher scores on decision making (Factor 1), collaboration (Factor 2), and nursing intervention (Factor 3), and these differences were significant (Factor 1: p < 0.001; Factor 2: p < 0.01; Factor 3: p < 0.01). Longer nursing experience in ICUs was associated with higher scores on all four competence areas, and these differences were significant (Factor 1: p < 0.001; Factor 2: p < 0.01; Factor 3: p < 0.01; Factor 4: p < 0.05). Nurses who had worked longer in ICUs and CNs in intensive care/CNSs in critical care nursing had higher scores on all four competence areas. ICU nurses with experience in emergency departments also showed significantly higher scores than did those without this experience (16.7 [SD 1.48] vs. 15.7 [SD 1.67], respectively, p < 0.001), but there was no significant difference in these scores between ICU nurses with and without experience in cardiovascular departments (16.4 [SD 1.80] vs. 15.9 [SD 1.64], respectively, p 0.08) or operating rooms (16.1 [SD 1.67] vs. 16.0 [SD 1.67], p 0.84).

## Discussion

In this study, we developed an original self-assessment questionnaire, the C3Q-safety, to measure areas of critical care nursing competence related to patient safety. Nursing competence, as

**Table 3. Factor structure of the C3Q-safety (n = 211).**

| Questionnaire Item | F1 | F2 | F3 | F4 | Cronbach's alpha |
|---|---|---|---|---|---|
| F1: Decision making (7 items) | | | | | 0.831 |
| 17 | 0.782 | | | | |
| 16 | 0.719 | | | | |
| 14 | 0.718 | | | | |
| 18 | 0.710 | | | | |
| 13 | 0.513 | | | | |
| 11 | 0.321 | | | | |
| 15 | 0.266 | | | | |
| F2: Collaboration (5 items) | | | | | 0.803 |
| 8 | | 0.805 | | | |
| 9 | | 0.711 | | | |
| 6 | | 0.643 | | | |
| 10 | | 0.424 | | | |
| 7 | | 0.379 | | | |
| F3: Nursing intervention (5 items) | | | | | 0.734 |
| 3 | | | 0.906 | | |
| 2 | | | 0.679 | | |
| 4 | | | 0.469 | | |
| 1 | | | 0.385 | | |
| 5 | | | 0.332 | | |
| F4: Principles of nursing care (5 items) | | | | | 0.795 |
| 20 | | | | 0.948 | |
| 19 | | | | 0.756 | |
| 22 | | | | 0.528 | |
| 21 | | | | 0.399 | |
| 23 | | | | 0.316 | |

C3Q-safety: Critical Care Nursing Competence Questionnaire for Patient Safety, F: factor

assessed by the C3Q-safety, consists of four factors: decision making, collaboration, nursing intervention, and principles of nursing care. Nurses licensed as CNs in intensive care/CNSs in critical care nursing and those with a long ICU experience received the highest scores on these areas of competence.

Undoubtedly, improving critical care nursing competence is important for patient safety in ICUs. To achieve this improvement, we first need to measure this competence. Three measurement scales—the Critical Care Competency Assessment, the Self-Assessment Competence Tool, and the ICCN-CS-1—have been published; however, these three scales include many

**Table 4. Correlation coefficients between factors.**

| | F1 | F2 | F3 | F4 |
|---|---|---|---|---|
| F1: Decision making (7 items) | – | | | |
| F2: Collaboration (5 items) | 0.72 | – | | |
| F3: Nursing intervention (5 items) | 0.61 | 0.53 | – | |
| F4: Principles of nursing care (5 items) | 0.62 | 0.56 | 0.47 | – |

F: factor

**Table 5. Comparisons based on characteristics and nursing competence areas.**

| Variable | | Total | F1: Decision making | F2: Collaboration | F3: Nursing intervention | F4: Principles of nursing care |
|---|---|---|---|---|---|---|
| Sex | | | | | | |
| | Male | 16.31 (1.86) | | | | |
| | Female | 15.95 (1.61) | | | | |
| | p | 0.20 | | | | |
| CN in intensive care/ CNS in critical care nursing | | | | | | |
| | Yes | 17.43 (1.60) | 4.34 (0.43) | 4.35 (0.51) | 4.40 (0.35) | 4.32 (0.50) |
| | No | 15.93 (1.63) | 3.92 (0.47) | 3.99 (0.52) | 4.04 (0.51) | 3.96 (0.49) |
| | p | 0.001 | 0.001 | 0.015 | 0.002 | 0.008 |
| Clinical ladder level | | | | | | |
| | 0 | 15.21 (1.56) | 3.70 (0.44) | 3.87 (0.54) | 3.77 (0.51) | 3.86 (0.49) |
| | 1 | 15.91 (1.58) | 3.92 (0.38) | 3.99 (0.51) | 4.01 (0.52) | 3.97 (0.48) |
| | 2 | 15.96 (1.61) | 3.89 (0.45) | 3.96 (0.54) | 4.11 (0.50) | 3.96 (0.51) |
| | 3 | 16.70 (1.72) | 4.18 (0.52) | 4.19 (0.51) | 4.22 (0.41) | 4.08 (0.53) |
| | 4 | 17.06 (1.39) | 4.21 (0.49) | 4.29 (0.47) | 4.42 (0.35) | 4.14 (0.45) |
| | p | < 0.001 | < 0.001 | 0.011 | < 0.001 | 0.18 |
| Total nursing experience (years) | | | | | | |
| | < 1 | 15.05 (1.52) | 3.66 (0.47) | 3.72 (0.55) | 3.78 (0.37) | 3.87 (0.49) |
| | ≥ 1, < 5 | 15.74 (1.50) | 3.82 (0.40) | 3.96 (0.48) | 3.97 (0.54) | 3.98 (0.47) |
| | ≥ 5, < 10 | 16.32 (1.74) | 4.08 (0.48) | 4.08 (0.56) | 4.17 (0.48) | 3.94 (0.54) |
| | ≥ 10 | 16.58 (1.71) | 4.13 (0.50) | 4.16 (0.52) | 4.21 (0.46) | 4.07 (0.50) |
| | p | 0.001 | < 0.001 | 0.008 | 0.002 | 0.39 |
| ICU nursing experience (years) | | | | | | |
| | < 1 | 15.19 (1.66) | 3.67 (0.51) | 3.72 (0.59) | 3.86 (0.53) | 3.92 (0.47) |
| | ≥ 1, < 5 | 15.97 (1.56) | 3.93 (0.43) | 4.03 (0.48) | 4.05 (0.50) | 3.95 (0.49) |
| | ≥ 5, < 10 | 16.86 (1.62) | 4.22 (0.44) | 4.19 (0.53) | 4.22 (0.47) | 4.13 (0.54) |
| | ≥ 10 | 17.50 (1.41) | 4.33 (0.47) | 4.41 (0.46) | 4.45 (0.31) | 4.29 (0.43) |
| | p | < 0.001 | < 0.001 | < 0.001 | 0.002 | 0.043 |

F: factor, CN: certified nurse, CNS: certified nurse specialist, ICU: intensive care unit

items and therefore require extensive time and effort to execute [8–10]. Further, asking large numbers of questions can lead to inaccurate answers. To solve these problems, we developed the C3Q-sefety, a 22-item questionnaire that participants can complete in about 5 minutes. The small number of questions makes it possible for participants to complete the questionnaire during their free time at work and facilitates repeating the measurement regularly. Future studies should use larger samples for the validation and revision of the C3Q-sefety.

Among the ICU nurses participating in the present study, those with more years of experience working as nurses and in ICUs scored higher on the C3Q-safety, compared with nurses with less experience. The results of multiple comparisons revealed that having more ICU experience was associated with significantly higher scores than was having more overall nursing experience. Additionally, CNs in intensive care/CNSs in critical care nursing had higher scores than non-certified nurses on all four competence areas. Thus, the C3Q-safety score can be considered to reflect a part of critical care nursing competence. Although we intend to develop more question items asking about patient safety, we have not yet been able to confirm whether

the C3Q-safety is actually correlated with patient safety in ICU settings. An instrument to assess patient safety, such as the Agency for Healthcare Research and Quality's Hospital Survey on Patient Safety Culture, could be used to verify the reliability and validity of the C3Q-safety [11].

We classified critical care nursing competence as assessed by the C3Q-safety using the ICCN-CS-1 as a reference. The ICCN-CS-1 is the most studied of the existing three tools to measure critical care nursing competence [12]. The ICCN-CS-1 uses 144 items to measure two basic competences: clinical and professional competence. Clinical competence can be classified into three domains: principles of nursing care, clinical guidelines, and nursing interventions. Professional competence can be classified into four domains: ethical activity and familiarity with health care laws, decision making, development work, and collaboration. In the present study, the four domains of the C3Q-safety revealed by the factor analysis were named principles of nursing care, nursing interventions, decision making, and collaboration, with reference to the seven domains of the ICCN-CS-1. The C3Q-safety's four domains were positively correlated with each other. In particular, decision making was highly correlated with the other three competence areas of collaboration (0.72), nursing interventions (0.61), and principles of nursing care (0.62; Table 4). The competence of decision making is presumed to be key among the four competence areas. This finding can be used for critical nursing education. By training nurses in decision making, the other three competences may also improve through synergy.

The competence of principles of nursing care, which is one of the clinical competence areas, did not show an obvious association with general nursing experience, such as years of experience working as a nurse or level on the clinical ladder. Because the principles of nursing care cover basic topics, even an unskilled nurse may be able to score fairly well on these items. We would like to consider modifying the content of these items to improve the power of detection.

We see a wide range of uses for the C3Q-safety. One is its use for education. The C3Q-safety is not merely intended to measure critical care competence; the purpose is to identify areas of poor competence and provide education based on the information, leading to improvements in the quality of medical care. Although this study shows that experience working in ICUs improves critical care nursing competence, we cannot wait years for nurses to master these areas. The C3Q-safety has the potential to solve this problem. ICU nurses with short work histories could be efficiently trained in a short period of time by providing high-quality experience (education) targeted to their weaknesses, instead of time-based experience. In addition to direct intervention for areas of poor competence, it would also be possible to indirectly improve poor competence areas through improving intervening competence areas that are positively correlated with other areas of competence.

This study had several limitations. First, the main limitation of the C3Q-safety is that it is based on self-assessment. People tend to self-assess their abilities favorably and to overestimate their abilities. This tendency is called the Dunning–Kruger effect, and it is more likely to occur among people with immature meta-cognitive ability and below-average ability in an area [13]. Actual ability might differ between the beginner and intermediate levels, even if scores on the C3Q-safety are the same.

Second, we could not verify the reason for CN/CNS's higher competence in this study. CNs/CNSs show higher competence, and there may be several reasons for their higher competence. The reasons are that they acquire competence for critically ill patients in the certified-nurse training, have years of experience, or generally have various nursing experiences. In other words, the higher competence of CN/CNS may be influenced not only by critical care certifications but also by multiple factors.

   

Third, we should mention limitation in reliability and validity of the C3Q-safety. Internal consistency reliability of the C3Q-safety was acceptable, because Cronbach's alpha coefficients varied between 0.73 and 0.83. Cronbach's alpha coefficient greater than 0.7 was considered to indicate acceptable reliability. However, the instrument's repeatability and reliability have not been assessed by test-retest analysis. Regarding the validity, we also need further verification. The C3Q-safety is composed of questions that are easy to understand, small in number, and well-reviewed by critical care experts, therefore, the content validity could be maintained to some extent. Although we have not assessed the external validity of the C3Q-safety using previous measurement scales, a higher score of the C3Q-safety depending on critical care certification and years of nursing experience is considered to indicate a certain degree of the external validity. In the future, the comparison of previous measurement scales with the C3Q-safety will reveal the C3Q-safety's characteristics and then lead to using these scales for different purposes in accordance to the goal.

Finally, this study had a small sample size and was conducted only in Japan. However, the sample size of the present study was by no means small compared with those used in previous studies, whose sample sizes range from 50 to 430 [4, 12]. Our data, collected from six ICUs with no regional bias in Japan, seem to accurately reflect the competence of ICU nurses in Japan. Nevertheless, it would be possible to increase the sample size and to administer the C3Q-safety globally in the future because the questionnaire can be completed in a few minutes. Further studies will lead to the verification and modification of the C3Q-safety.

## Conclusions

We developed an easy-to-use questionnaire to measure nursing competence related to patient safety in ICUs. The C3Q-safety was able to detect four factors: decision making (seven items), collaboration (five items), nursing intervention (five items), and principles of nursing care (five items). The C3Q-safety will make it possible to easily measure critical care nursing competence and can also be utilized for efficient education. Further verification and adjustment of the instrument should be conducted.

## Supporting information

**S1 Table. Raw data of C3Q-safety.**
(XLSX)

## Acknowledgments

We thank all of the intensive care unit nurses who participated in this study. We would also like to thank Yoko Nishio, Kumi Kawabe, Chiemi Kirimoto, Ayuko Anzai, Takumi Unten, and Makoto Igarashi for their contributions to this work. We thank Jennifer Barrett, PhD, from Edanz Group (www.edanzediting.com/ac) for editing a draft of this manuscript.

## Author Contributions

**Conceptualization:** Masatoshi Okumura, Tomonori Ishigaki.

**Data curation:** Masatoshi Okumura.

**Formal analysis:** Masatoshi Okumura.

**Investigation:** Masatoshi Okumura, Kazunao Mori.

**Methodology:** Tomonori Ishigaki.

**Project administration:** Masatoshi Okumura.

**Supervision:** Yoshihiro Fujiwara.

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
