## [Decision Letter · Decision Letter 0]

19 Sep 2019

PONE-D-19-19921

Development of an easy-to-use questionnaire assessing critical care nursing competence in Japan: a cross-sectional study

PLOS ONE

Dear Prof Okumura,

Thank you for submitting your manuscript to PLOS ONE. After careful consideration, we feel that it has merit but does not fully meet PLOS ONE’s publication criteria as it currently stands. Therefore, we invite you to submit a revised version of the manuscript that addresses the points raised during the review process.

Please see comments below. 

We would appreciate receiving your revised manuscript by 18 October 2019. To enhance the reproducibility of your results, we recommend that if applicable you deposit your laboratory protocols in protocols.io, where a protocol can be assigned its own identifier (DOI) such that it can be cited independently in the future. For instructions see: http://journals.plos.org/plosone/s/submission-guidelines#loc-laboratory-protocols

We look forward to receiving your revised manuscript.

Kind regards,

Andrew Soundy

Academic Editor

PLOS ONE

Journal Requirements:

Additional Editor Comments:

Please see comments from the reviewer and justify your response.

From looking at the manuscript please expand your methods section to include all required sections from this type of study: see https://www.strobe-statement.org/index.php?id=available-checklists

Reviewers' comments:

Reviewer's Responses to Questions

**Comments to the Author**

1. Is the manuscript technically sound, and do the data support the conclusions?

Reviewer #1: Partly

2. Has the statistical analysis been performed appropriately and rigorously? 

Reviewer #1: Yes

3. Have the authors made all data underlying the findings in their manuscript fully available?

Reviewer #1: Yes

4. Is the manuscript presented in an intelligible fashion and written in standard English?

Reviewer #1: Yes

5. Review Comments to the Author

Reviewer #1: Thank you for your desire and effort to advance nursing science. I appreciate the effort required to completed this study and develop the manuscript. The competence for critical care nursing is a very important topic that warrants ongoing attention by nurse scholars.

My comments are intended to help guide revisions.

- Introduction

Is the number of the certified nurses (in intensive care) correct ? According to the reference, it is described as 1033.

The CNS (critical care nursing) is not mentioned throughout in this paper. I think the certified nurse specialists work at ICU, but why is it excluded?

- Material and Methods

In the process of preparing the questionnaire, verification of reliability and validity is necessary. I think stability verification is necessary, because this questionnaire assesses competency. If not, I think you should add to the limits of the research.

What is the cut point of factor loading in exploratory factor analysis? The factor loading of item No, 5, 11, 15, and 23 seem to be low. Why these items are adapted?

There is no description of the results of confirmatory factor analysis (model fit or coefficient). If verified, description is required.

-Discussion

The competence is compared by multiple back grounds of nurses. Certified nurses are included in comparison of years of experience. The years of experience of certified nurses are unknown. I can’t recognize whether the influence of competence is due to a certified license or years of experience.

6. PLOS authors have the option to publish the peer review history of their article (what does this mean?). If published, this will include your full peer review and any attached files.

Reviewer #1: No

---

## [Author Response · Author response to Decision Letter 0]

10 Oct 2019

Dear Dr. Soundy:

Thank you for your ongoing consideration of our manuscript (PONE-D-19-19921) for publication in the PLOS ONE. We appreciate the time spent by you and the reviewer and believe the revised manuscript is improved. We hope that the revised manuscript would be now suitable for the publication.

We look forward to hearing from you regarding our submission. We would be glad to respond to any further questions and comments that you may have.

Sincerely,

Masatoshi Okumura

---

## [Decision Letter · Decision Letter 1]

11 Nov 2019

Development of an easy-to-use questionnaire assessing critical care nursing competence in Japan: a cross-sectional study

PONE-D-19-19921R1

Dear Dr. Okumura,

We are pleased to inform you that your manuscript has been judged scientifically suitable for publication and will be formally accepted for publication once it complies with all outstanding technical requirements.

With kind regards,

Andrew Soundy

Academic Editor

PLOS ONE

Additional Editor Comments (optional):

Reviewers' comments:

Reviewer's Responses to Questions

**Comments to the Author**

1. If the authors have adequately addressed your comments raised in a previous round of review and you feel that this manuscript is now acceptable for publication, you may indicate that here to bypass the “Comments to the Author” section, enter your conflict of interest statement in the “Confidential to Editor” section, and submit your "Accept" recommendation.

Reviewer #1: All comments have been addressed

2. Is the manuscript technically sound, and do the data support the conclusions?

Reviewer #1: Yes

3. Has the statistical analysis been performed appropriately and rigorously? 

Reviewer #1: Yes

4. Have the authors made all data underlying the findings in their manuscript fully available?

Reviewer #1: Yes

5. Is the manuscript presented in an intelligible fashion and written in standard English?

Reviewer #1: Yes

6. Review Comments to the Author

Reviewer #1: (No Response)

7. PLOS authors have the option to publish the peer review history of their article (what does this mean?). If published, this will include your full peer review and any attached files.

Reviewer #1: No

---

## [Editor Report · Acceptance letter]

15 Nov 2019

PONE-D-19-19921R1 

Development of an easy-to-use questionnaire assessing critical care nursing competence in Japan: a cross-sectional study 

Dear Dr. Okumura:

I am pleased to inform you that your manuscript has been deemed suitable for publication in PLOS ONE. Congratulations! Your manuscript is now with our production department. 

With kind regards,

on behalf of

Dr. Andrew Soundy 

Academic Editor

PLOS ONE